# Simulation of the Grondona System of Conditional Currency Convertibility Based on Primary Commodities, Considered as a Means to Resist Currency Crises

**Patrick Collins** [1,*,†], **Jameel Ahmed** [2] **and Ahamed Kameel Meera** [3]

1   Azabu University, Sagamihara City 252-5201, Japan
2   Institute of Management Sciences, University of Baluchistan, Quetta 87300, Pakistan;
    jamil.ahmed@um.uob.edu.pk
3   Department of Finance, International Islamic University of Malaysia, Kuala Lumpur 68100, Malaysia;
    akameel@gmail.com
*   Correspondence: collins@azabu-u.ac.jp
†   Retired from the Azabu University.

**Abstract:** Currency crises are a significant feature of the present-day world economy, in which financial transactions are many times larger than monetary flows in the "real economy", so that defending a currency's exchange-rate is a major challenge for the governments of countries which may be smaller than a single large corporation. It is made even more difficult due to the United States government and its agents openly using economic pressures to try to force other countries to obey its orders, even including regime change. Guaranteed convertibility of a currency, such as maintaining a gold standard, can in principle help to stabilise its value, but this has been absent since the end of US dollar convertibility in 1971. The Grondona system of conditional currency convertibility was not planned as a counter-measure for currency crises. However the simulation of its operation demonstrated in this paper shows clearly how its automatic counter-cyclical stock-holding in response to movements in commodity prices—and so to exchange-rate movements that alter domestic commodity prices—causes monetary flows that would resist large exchange-rate movements (among other effects), and thereby tend to ameliorate a currency crisis. Moreover, it would achieve this without the need for international negotiations, agreements or other geopolitical trade-offs.

**Keywords:** Grondona system; currency convertibility; commodity price stabilisation; currency crisis

## 1. Introduction: The Challenge of Currency Crises

Perhaps the most visible and best-known aspect of a currency crisis is a sharp fall in a currency's exchange-rate, which may be either the result or the cause of crisis. In today's world of countries of very different sizes linked by trade and financial markets, a fall in the exchange-rate may be caused either wittingly or unwittingly by the actions of a single large organisation, whether a country's government or a large corporation, as well as by loss of international confidence in the government's monetary policy.

A large and/or sudden drop in the exchange-rate puts a burden on domestic importers, leads to imported inflation, and makes foreign debt repayments heavier. Government's counter-measures may also impose severe costs on a country. For example, a government may sell foreign exchange reserves, or may raise interest-rates to strengthen the currency and resist inflation, thereby causing a severe deceleration in domestic economic activity. The risk of currency crisis increases as international

financial markets widen and deepen, thereby becoming more complex, including due to international derivative market operations, of which the scale dwarfs the real economy today.

The problem is well exemplified by the South-East Asian currency crisis of 1997–1998 which exhibited many of the features that make a currency crisis so challenging for policy makers: contagion from one country to another, high speed of events, high cost of policy-mistakes, limited capabilities of policy instruments, vulnerability to external pressure—all aggravated by a high level of uncertainty. The importance of high uncertainty is well shown by the case of Malaysia during that crisis: in the face of a tidal wave of severe criticism from Wall Street, the US government and US news media, Prime Minister Mahathir stood his ground and adopted a policy that was recognised, several years later, to have been by far the most successful! This event was a profound—though largely ignored—demonstration of the very limited value of "mainstream economics" for policy-making: if the advice offered at such a critical time by an overwhelming majority of "respected" economists, policy-makers and economic commentators in the mainstream media, dominated by the USA, is so poor, what is these peoples' function? And it raises the age-old question: "Do they know that what they are saying is wrong? That is, that their advice would actually aggravate the problem?" If they do not know, then they are incompetent, and should not be in the influential position they have; but if they do know, then they are dishonest—literally enemies pretending to be friends.

Much the same criticism can be made concerning the 2008 financial crisis spreading from Wall Street and causing recession around the world, nominally triggered by the collapse of Lehman Brothers Inc. Did the thousands of well-paid economics professors in American universities not know how extremely fragile the US financial system was? Or did they know but kept silent? Is it actually a condition of economists' employment that they must not discuss the fundamental instability—and dishonesty—of the US fractional-reserve banking system? Solving this ever-worsening problem is the central goal of the work of the Public Banking Institute (PBI) and American Monetary Institute in the USA, among other organisations.

There is an extensive literature on currency crises, including various generations of currency crisis models which discuss the causes of crises and political remedies. In parallel, a range of strategies have evolved over recent years to mitigate the risk of currency crisis: building up foreign exchange reserves commensurate to a country's holdings of foreign debt, making currency swap arrangements with allied countries, use of "derivatives" such as currency futures, and others.

An additional aspect that must be considered is political risk. Although a country's government and corporations may be following normally adequate prudential policies, it may be targeted for destabilisation for political reasons by the US government. As of 2019, this is being done openly against Iran, Russia, Syria, North Korea and Venezuela, among other countries which are the target of publicly announced economic attack by the US government, often intended to achieve "regime change". Defence against such politically motivated economic pressure is more complex and needs a stronger basis than defence against purely economic risks. That is, while the US government continues to use its dominant role in dollar-based economic activities to try to force other countries to follow its orders, relatively smaller countries in particular need to try to insulate themselves from such political pressures, in addition to the risks and problems caused by major market fluctuations.

A counter-measure that has been spreading in recent years is for countries to avoid using the US dollar in their trade. This is implemented through bilateral agreements between pairs of countries to make arrangements to pay for mutual imports and exports in their own currencies. In order for such agreements to be successful, it is necessary to achieve a satisfactory level of stability in the relative value of their respective currencies. However, due to the absence of any component of convertibility of currencies into real commodities today, the value of a country's currency is defined solely by activity in foreign exchange markets. This dependence on foreign exchange markets makes the currency of all but the largest countries subject to manipulation by large operators, both commercial and governmental.

The present paper takes a different approach: it is concerned neither with the causes of crises, nor with the politics of counter-measures. It considers the extent to which an "automatic stabilizer"

(i.e., driven by market prices), which operates predictably, regardless of the causes of a crisis, whether political or economic, can reliably exert a stabilizing influence on various economic parameters relevant to overcoming a currency crisis. Unemployment insurance has been called a macro-economic "automatic stabilizer", since it automatically reduces the extent to which consumption falls as a result of rising unemployment, analogous to the stabilisers on a ship: whatever the cause of turbulence in the sea, whether wind, earthquake, typhoon or another cause, they automatically reduce the extent to which the ship rolls from side-to-side. Likewise, unemployment insurance acts dependably, on a scale decided in advance by the government, to prevent the vicious circle by which recession worsens into depression, without giving rise to suspicion of political misjudgement or nepotism, nor being vulnerable to distortion or failure caused by speculative attack.

Another example, which has been known for centuries but is not widely discussed today, is the "automatic" macro-economic stabilizing effects of a system of currency convertibility based on primary commodities. Because of inadequacies in the operation of gold convertibility, even during the 19th century heyday of the gold standard, convertibility based on a range of basic commodities was recommended by the first of a long list of influential economists, including (Jevons 1877; Marshall 1887; Fisher 1913; Fisher 1928; Graham 1937; Keynes 1938; Hayek 1943; Hart et al. 1964; Riboud 1981; Kaldor 1976; Luke 1975; Kaldor 1983; Borsodi 1989; Greco 1990; Lietaer 2001; Ussher 2011; Ussher 2012) and others from across the political spectrum. None of these plans for commodity convertibility have been implemented, because they were either impractical (that is they would not be able to achieve their stated objective, such as to keep commodity prices between fixed limits), or would have required detailed international negotiations among multiple countries for their possible implementation for which there has not been the required political will. Further, most of these commodity-backed currency-systems were described only conceptually without providing detailed guidelines for implementation, nor for how to perform simulations of their operation in order to analyse their possible effects.

This long-term and widespread support has not attracted the attention that might be expected. As a single striking example, the recent book "Keynes Hayek: The Clash that Defined Modern Economics" (Wapshott 2012), of which the theme is representative of present-day "mainstream" economic thinking, *does not even mention* the fact that, far from being entirely opposed in their thinking, both Keynes and Hayek wrote strongly in favour of the macro-economic stabilizing benefits of a system of currency convertibility based on primary commodities, due to its automatic counter-cyclical stock-holding function. Their strikingly similar views are epitomised in the following short excerpts from their writings on the subject:

> "At present a falling off in effective demand in the industrial consuming countries causes a price collapse . . . But if . . . "Commodity Controls" are in a position to take up at stable prices the slack caused by the initial falling off in consuming demand . . . the vicious cycle may be inhibited at the start; and, again, by releasing stocks when consumption recovers . . . prevent the inflation of raw material prices.". (Keynes 1938)

> "With this system in operation an increase in the demand for liquid assets would lead to the accumulation of stocks of raw materials of the most general usefulness . . . And as the hoarded currency was again returned to circulation and demand for commodities increased, these stocks would be released to satisfy the new demand.". (Hayek 1943)

These two short excerpts do not discuss another inherent aspect of the operation of such a system, namely that it would also cause a limited, counter-cyclical variation in the supply of the currency in which it operated. As a result, the system would also act indirectly to resist changes in the exchange-rate and both inflation and deflation, like an "automatic stabiliser". However, despite this endorsement at literally the highest level of the mainstream "western" economics establishment, such a system has never been implemented, and indeed, devising a practical plan is not easy, although the difficulty is rather different from that of resuming gold convertibility.

In the case of gold, proposals to revive an international gold standard face such problems as the need to revalue gold by a factor of approximately 10× in order to provide cover for world currencies, which would cause gross economic injustice due to the very uneven ownership of gold. In addition, the system would be vulnerable to speculative attack, not least by geopolitical competitors, who have essentially unlimited "fiat" funds at their disposal. Consequently, a realistic attempt to implement a modern gold standard would require international agreement—which removes it from consideration as a "practical" measure such as a sovereign national government can implement independently.

Instead, the key problem faced in trying to implement currency convertibility based on primary commodities is that it must accommodate continuing wide movements in their market prices, which are essential for markets to balance demand and supply. This problem is much less significant for gold, since its industrial uses are far less than its use as a store of value. By contrast, most primary commodities are important components of international trade and industry, and their market prices are notoriously unstable, swinging widely over the business cycle by as much as −50% and +100% or more. Trying to use commodities with such widely moving prices as the basis of currency convertibility seems, prima facie, self-contradictory. However, if it were possible, even partially stabilising primary commodity prices would have a beneficial stabilizing effect on world trade and economic growth, for which reason both Keynes and Hayek strongly supported the principle, rather than trying to revive a gold standard, which would have no direct stabilising influence on world commodity trade and markets.

However, the various international plans that have been proposed to implement the concept of commodity convertibility have unfortunately all been demonstrably impractical—requiring an unrealistic degree of international cooperation, an unacceptable level of interference with market forces, excessive dependence on experts' discretionary judgement, lack of transparency, or other critical problems, as discussed further in (Collins 1985).

The main contribution of the present paper is to provide evidence to justify further attention to the system described in the following section, which the authors propose is a genuinely practical means of implementing currency convertibility based on primary commodities, able to be safely implemented by individual countries, and thereby offering a useful, incremental step towards strengthening them against currency crises. An additional strength of the system is that its operation can be simulated reliably. The current paper illustrates this in the following sections by describing the Grondona system, simulating its operation in Indonesia, a D-8 member country, and thereby showing its resulting economic effects. Finally, it discusses their economic implications, and the potential value of wider implementation by D-8 countries.

## 2. Grondona System of *Conditional* Currency Convertibility

In contrast to the various international systems proposed over more than a century, the Australian writer Leo St. Clare Grondona (1890–1982) developed a less ambitious system of conditional currency convertibility during the 1950s, comprising a multi-polar system which enables individual countries to partially stabilise their currencies in real terms, independently, on a scale determined in advance. Most importantly, Grondona prepared a detailed plan of how to implement this form of partial price stabilisation of durable, essential, basic, imported commodities, which is also a system of conditional currency convertibility based on these primary commodities (Grondona 1975).

NB Grondona stipulated that only durable, essential, basic, imported commodities should be included, and handled only in large units of quantity, in order to minimise operating costs and avoid problems of deterioration, as well as political issues of subsidising domestic primary production, which might arise with products for which the country did not depend on imports.

Studies which evaluated the Grondona system from Shariah perspective, and which concluded that its operations are indeed Shariah-compliant, have been published in (Ahmed et al. 2014; Ahmed 2015; Ahmed et al. 2018b), and on-line at https://emeraldinsight.com/doi/abs/10.1108/JIABR-05-2015-0018?journalCode=jiabr. In outline, the Grondona system of currency convertibility is based on the

fundamental principle of economic planning of Prophet Yusuf (AS), accumulating reserves of primary commodities during times of abundance and releasing those reserves during periods of scarcity, as described in Holy Quran (Ahmed 2015). Briefly, the currency that is "created" to pay for the CRD's reserves is backed by the real commodity reserves which it is used to purchase. As the quantity of reserves falls (when market prices rise) the currency is retired from circulation (at a pre-announced premium). Hence, even under an existing fiat money system, the CRD's role cannot be criticized as "riba", neither for issuing "fiat" money, nor for being debt-based, like all currencies which are largely supplied via the fractional-reserve banking system. This is a very encouraging result, since it means that implementing the Grondona system in terms of their domestic currencies could be a way for D-8 countries to improve their macro-economic stability in the face of destabilising external influences.

Grondona described his system of partial price stabilisation of primary commodities in detail in many speeches, articles and papers, starting from 1950 (Grondona 1950), and in a series of books (Grondona 1958; Grondona 1962; Grondona 1964; Grondona 1972; Grondona 1975). It is not possible to summarise all of this information in a single paper. However, the underlying idea is very simple, but in critical points it is different from related proposals, and so the effects of its operation are very different. In particular, the financial liability involved in implementing the system is limited in advance by the government establishing it, thereby avoiding the open-ended liability involved in commodity "buffer-stock" schemes, which attempt to limit the movement of commodity prices. This has the important implication that, in contrast to other proposals for commodity-backed currency, individual countries are able to implement the Grondona system independently in terms of their own currencies, legitimately paying for purchases of reserves through monetary expansion rather than taxation, as was done under the gold standard. Grondona envisaged that, in the long term, many countries would adopt the system, thereby leading to greater stability of primary commodity prices, and also indirectly to gradual stabilisation of exchange-rates, in addition to smoothing the trade cycle. In this way, the Grondona system avoids the extremely difficult and unpredictable process of trying to negotiate different countries' shares of an international system, while preserving each country's sovereignty and ensuring mutual benefits from implementation. This is a particularly important advantage from the perspective of political feasibility or "practicality".

A second critical difference from other related proposals is that the guarantee to provide commodities in exchange for monetary units on demand at specified prices would apply only as long as the system was holding reserves of those commodities (the exact levels of which would be publicised at all times). Thus, on occasions, reserves of one or more commodities might fall to zero, making the system's guarantee of a minimal value of the currency in terms of those commodities temporarily ineffective. However, the system would continue to provide support to the commodities' market prices, and reserves of the commodity would be likely to subsequently accumulate as market prices fell once again, making the system's maximum prices effective once more. This and other important details of Grondona's proposals are described at length in (Grondona 1958; Grondona 1975; Collins 1985; Ahmed 2015), and on-line at https://link.springer.com/book/10.1007/978-1-349-07058-9.

A notable advantage of the Grondona system over the gold standard is that it is fully counter-cyclical: that is, it is automatically stimulatory as well as contractionary over the business cycle. It was a major weakness of the gold standard that it obliged countries with a trade deficit and/or inflation to contract demand, but did not oblige surplus countries to expand demand. As a result, it had a net deflationary tendency, leading to unnecessarily high unemployment and slow economic growth. This is a serious weakness also of the operation of the IMF which remains from the Bretton Woods gold exchange standard system, as discussed at length by (Stiglitz 2002) and (Pettifor 2003). By contrast, under the Grondona system a fall in commodity prices in the domestic currency, and/or a rise in the exchange-rate, would automatically expand the money supply and the flow of currency abroad, thereby stimulating economic activity automatically in response to deflationary market pressure as advocated (as noted above) by both (Keynes 1938; Hayek 1943).

Institutionally, the Grondona system would be implemented by a "Commodities Reserve Department" (CRD) which would stand ready to purchase or sell reserves of specified commodities at prices in its official "price schedules" on demand (as illustrated below). It would not actively enter the market but would respond predictably to market participants who wished to sell or purchase commodities from its reserves on its published terms. Being passive it would not try to alter market prices, but its predictability would provide a valuable link between the monetary world and the real economy, while being compatible with a country's existing monetary system, using the national currency.

## 2.1. Details of Implementation

For each of the durable, essential, basic, imported commodities involved, the CRD would publish a "price schedule", according to which the prices paid or accepted by the CRD in exchange for specified (large) units of that commodity would adjust in proportion to the CRD's current level of reserves of the commodity, as illustrated in Table 1 for the case of Aluminium, in Indonesian Rupiah. This is not a plan: it merely shows that, having initially offered to pay 83,389,528 Rupiah per ton of Aluminium (of a standard grade used in world markets), if the CRD's reserves reach a level of 6900 tons, then the price which the CRD will subsequently offer will fall to 79,220,056 Rupiah per ton. The CRD's offer price and sale price (which Grondona called its "low point" and "high point" would continue to adjust according to this "price-schedule" as the quantity of its reserves rose or fell.

**Table 1.** Illustrative Rupiah Price-Schedule for Aluminium.

| Current CRD Buying Price (Low Point) Rp/Tonne | Current CRD Selling Price (High Point) Rp/Tonne | Max. Quantity in Indonesian CRD's Reserves | Number of Blocks |
|---|---|---|---|
| 83,389,528 | 101,920,536 | 6900 | 1 |
| 79,220,056 | 96,824,512 | 13,800 | 2 |
| 75,050,576 | 91,728,480 | 20,700 | 3 |
| 70,881,096 | 86,632,456 | 27,600 | 4 |
| 66,711,624 | 81,536,432 | 34,500 | 5 |
| 62,542,148 | 76,440,400 | 41,400 | 6 |
| 58,372,672 | 71,344,376 | 48,300 | 7 |
| 54,203,192 | 66,248,348 | 55,200 | 8 |
| 50,033,716 | 61,152,320 | 62,100 | 9 |
| 45,864,240 | 56,056,296 | 69,000 | 10 |
| 41,694,764 | 50,960,268 | 75,900 | 11 |
| 37,525,288 | 45,864,240 | 82,800 | 12 |
| 33,355,812 | 40,768,216 | 89,700 | 13 |
| 29,186,336 | 35,672,188 | 96,600 | 14 |
| 25,016,858 | 30,576,160 | 103,500 | 15 |

More generally, with a CRD in operation, when a commodity's market price in terms of that country's currency was falling (that is, when the value of that country's monetary unit in terms of a particular commodity was rising), market participants would sell commodities to the CRD in exchange for monetary units once the CRD's current buying price became attractive relative to current market prices. When the quantity of reserves of the commodity rose to a pre-specified quantity (which Grondona termed a "Block"), the CRD's official buying and selling prices for that commodity (known as its "points") would fall by a pre-specified amount. If the value of the monetary unit in terms of this commodity continued to rise, so that market prices fell to this new, lower "point" and reserves continued to accumulate, the process would repeat, and the cycle would continue until the CRD's buying price (lower "point") fell low enough to be unattractive to sellers. Later, when the value of the monetary unit in terms of that commodity declined as market prices recovered, buyers would repurchase reserves from the CRD at the successively higher selling prices (upper "points") in its published reserve price schedule, as each in turn became attractive relative to the current market price.

In the following it is assumed that an Indonesian CRD is established following Grondona's guidelines (Grondona 1975). As described above, the Indonesian Rupiah would thereby become

convertible into a range of durable, essential, basic, imported commodities, at prices which would adjust according to the level of reserves held by the CRD, following each commodity's published "price schedule".

The simplicity of the Grondona system also enables detailed simulations of its operation in order to examine its potential effects under different conditions, without the need to use a macro-economic model to evaluate its impact (although such could be used in more advanced simulations). Collins simulated the timing and scale of changes in the national money supply that would have occurred in Japan during the 1990s, on the assumption that CRDs were established on representative terms (Collins 1996a, 1996b). Ahmed performed simulations of the Grondona system in the four D-8 countries Pakistan, Malaysia, Turkey and Indonesia in (Ahmed 2015). Recently, Ahmed, Collins and Meera performed more up-to-date simulations of the Grondona system for Turkey and examined its monetary effects (Ahmed et al. 2018a).

Because of its "automatic" operation, activated by market forces rather than by political or "expert" judgement, the Grondona system can be realistically simulated with a high degree of confidence. Consequently, these simulations of its potential operation in different countries have helped to illustrate the counter-cyclical timing of its automatic functioning (Ahmed 2015).

The development of these simulations has also created an essential tool for the government of any country either considering or planning to implement the system: in order to decide the optimal initial conditions (including which commodities and grade(s) to include, the maximum quantity of reserves to hold at each price level, the initial price-levels and the steps in the price-schedule, for each commodity), being able to run numerous simulations based on a range of different initial conditions is clearly very valuable.

## 2.2. Costs and Risks

The main cost of implementing the Grondona system comprises the cost of preparing and maintaining the warehousing needed for the commodities that will be accumulated by the CRD. Grondona discussed this in detail, including such details as that CRD warehouses should be sited near ports and/or major users within the host country (Grondona 1975). This cost is determined by the scale on which the system is set up, and so will be chosen not to be an excessive burden on the government budget.

A second "cost" is that of permitting the money supply to increase and decrease in proportion to the CRD's reserves (within predetermined limits). That is, as under the gold standard, it is an important aspect of the CRD's operation that the funds used to pay for the reserves should not be raised from taxation or government borrowing but should comprise "new money" released into the economy. Likewise, the proceeds of sales of the CRD's reserves should not be treated as government revenues but should leave circulation. This makes the reserves "costless" (subject to the obvious condition that the scale of implementation is not excessively large, thereby causing severe distortion of the money supply). As discussed in detail in (Collins 1985), counter-cyclical variation in the money supply will in general be beneficial: however, it can be countered by the monetary authorities, if desired, without cancelling the beneficial stabilising influence on each commodity trade and industry, due to the different route by which monetary policy acts on the economy (such as through government bond market operations). A priori it is equally likely that the monetary authorities would wish to accentuate the monetary effects of the CRD operations, due to their counter-cyclical timing, as discussed in (Collins 1985).

Considering the potential risks of "political economy" that are involved in all government activities, decisions on the siting of the CRD's warehousing may be used for political purposes of regional development, use of favoured contractors, and/or other political purposes, as is common with public works projects. Deciding the scale of the system's operation in respect to different commodities may also be distorted in favour of some commodities. However, provided that the central principle of the CRD's automatically adjusting support-prices is preserved, there are no complex macro-economic or geo-political issues, such as politically controversial trade-offs typical of international negotiations that need to be resolved. Due to this, the authors consider it appropriate to describe the system as "politically practical".

Among other risks, the Grondona system is not vulnerable to attack by speculators, due to the predictable conditionality of its stabilizing operations. For example, a "speculative attack" on a CRD might involve buying all its reserves of one or more commodities, or conversely selling large quantities of commodities to it. But in either case the CRD would benefit—either by selling all its reserves at some 20% above the prices which it paid (as per its price-schedules), or by accumulating reserves of essential imports at ever-lower prices. Hence, except in case of actual fraud, the CRD is literally immune to speculative attack—while at the same time acting to strengthen the currency against speculative attack in the foreign exchange markets, due to its counter-cyclical response to major changes in commodity market prices (i.e., in its national currency) that may be caused by large exchange-rate movements.

Nor would a CRD be vulnerable to harm from "competition" from another country's CRD. Any "competition" between different countries' CRDs would comprise their setting higher prices in order to attract reserves. However, provided that they use the central feature of the Grondona system, namely the market price-driven adjustment of its support prices in its price-schedules, their buying prices would adjust downwards after accumulating a certain quantity of reserves, thereby making the price offered by other countries' CRDs more attractive (allowing for considerations of exchange-rate risk). Hence, there would be no need for coordination between different countries establishing CRDs, and any "competition" between different countries' CRDs would act to improve the stability of commodity prices and trade by increasing the quantity of counter-cyclical stock-holding capacity in the world economy.

## 3. Simulation of Grondona System Operation in Indonesia

The following case study of Indonesia, selected in view of its role as a member of the "D-8" group of leading Islamic countries, illustrates how the automatic adjustment mechanism of the Grondona system exerts a stabilising influence on the real value of the currency of a country which implements the system. This stabilising influence comprises direct stabilisation in terms of the commodities handled, and indirect stabilisation through counter-cyclical changes in the money supply. The simulations are based on the principles and guidelines suggested in (Grondona 1975) and use past data on the annual quantities and values of Indonesia's imports of primary commodities, and their monthly market prices, found in World Integrated Trade Solution (WITS) on the IndexMundi website. The simulation results show how the Indonesian money supply changes in parallel with changes in the levels of reserves of the different primary commodities stockpiled by the Indonesian CRD. By this, the simulation results show how the automatic price adjustment mechanism of the Grondona System, by partially stabilizing the real value of the Rupiah, would thereby help to insulate the Indonesian economy from fluctuations of the business cycle and other external shocks.

Grondona suggested that, in practice, each country should decide the details of the initial conditions of implementation to suit their own conditions, but he himself offered some preliminary guidelines. For simplicity, in the following we use the uniform guidelines suggested in (Grondona 1975):

i.      The CRD would make only Indonesian Rupiah transactions and would operate without national discrimination.

ii.     It would handle only selected commodities of specified standard grades, in specified, large units of quantity. The CRD would have no dealings with commodity futures, currency or financial markets.

iii.    It is also assumed that Indonesian imports would continue to come from the same supplying countries as before the CRD was established, and that domestic users of the commodities in question would continue to use approximately the same quantities as they did before the CRD was established.

### 3.1. Simulation Methodology

The simulation of an Indonesian CRD was performed by using past data on the annual quantities and values of Indonesia's imports of primary commodities, and their monthly average market prices,

using a program developed in C++ to perform the simulations. The outputs of the simulations were further analysed in Microsoft Excel to generate relevant graphs.

### 3.2. Data Description

Indonesian primary imported commodities were selected based on the attributes recommended in (Grondona 1975), namely durable, essential and basic. Based on these attributes, the authors used the Harmonized System (HS) 6-digit codes to select a list of primary commodities imported by Indonesia, and used the 6-digit HS codes to retrieve their annual quantities and trade values from the World Integrated Trade Solution (WITS) developed by the World Bank (World Bank 2013). The HS 6-digit codes provide more detailed information about trade statistics than other nomenclatures, and they have been used since 1988. Table 2 shows the HS 6-digit codes for Indonesian primary commodity imports. The HS codes of imported primary commodities of Indonesia listed in Table 2 were used to identify the required primary commodities from the full list of imported commodities.

**Table 2.** Country-Wise List of Primary Commodities with HS Codes.

| S. No | Product Description | HS Product Codes |
|-------|---------------------|------------------|
| 1. | COFFEE NOT ROAST, NOT DECAFEINATED | 090111 |
| 2. | DURUM WHEAT | 100110 |
| 3. | BARLEY | 100300 |
| 4. | SOYA BEANS, WHETHER OR NOT BROKEN | 120100 |
| 5. | COCOA BEANS, WHOLE, BROKEN, RAW OR ROAST | 180100 |
| 6. | RAW SUGAR, NOT CONTAINING ADDED FLAVOURING OR COLOURING MATTER: CANE SUGAR | 170111 |
| 7. | COTTON, NOT CARDED OR COMBED | 520100 |
| 8. | RICE IN HUSK (PADDY OR ROUGH) | 100610 |
| 9. | JUTE AND OTHER TEXTILE BAST FIBRES, RAW OR RETTED | 530310 |
| 10. | NATURAL RUBBER LATEX, WHETHER OR NOT PRE-VULCANISED | 400110 |
| 11. | REFINED COPPER, CATHODES & SECTIONS | 740311 |
| 12. | NICKEL, NOT ALLOYED | 750210 |
| 13. | ALUMINIUM, NOT ALLOYED | 760110 |
| 14. | REFINED LEAD | 780110 |
| 15. | ZINC CONTAINED BY WT>99.99% NOT ALLOYED | 790111 |
| 16. | ZINC CONTAINED BY WT<99.99% NOT ALLOYED | 790112 |
| 17. | TIN, NOT ALLOYED | 800110 |

The annual quantities and trade values, obtained from WITS for Indonesia, were given in kilograms and U.S. dollars respectively. Thus, the authors converted the annual quantities into tonnes, and trade values into the domestic currency of Indonesia i.e., Indonesian Rupiah. For conversion of trade values, the annual exchange-rates were obtained from the World Bank website. Annual data on Indonesia's Consumer Price Index (CPI) were also retrieved from the database of the World Bank, in order to adjust the prices of primary imported commodities for inflation as suggested in (Grondona 1975).

Table 2 shows the complete list of candidates among Indonesia's primary imported commodities. After initial examination of these data, the authors found that some of the commodities are imported either in small quantities or have missing values for a few years. Thus, such commodities were excluded from the simulation. Table 3 represents the final list of primary imported commodities which were included in the simulation.

The monthly market prices of the above listed Indonesian primary commodities were retrieved from the IndexMundi website. This monthly data was available in the domestic currency of Indonesia (IDR) on the given website. Since the data about a few commodities, namely Cocoa Beans, Sugar,

Cotton, Coffee and Rubber were given in IDR/KG, the authors converted them into IDR/Tonne by multiplying each month's price by 1000.

There are five parameters (which Grondona called the "gearing" of the system), which determine the extent of the system's monetary and economic influence, and the government's financial commitment involved in implementing this system. These parameters are the range of commodities, the initial price level for each commodity, the size of "Blocks", the width of the price-band for each commodity, and the price steps between successive price-bands.

**Table 3.** List of Primary Imported Commodities Selected for Simulation.

| Country | Agricultural Commodities | Metals | Total Primary Commodities |
|---|---|---|---|
| Indonesia | Soybean; Coffee; Cocoa Beans; Sugar; Cotton; Rice in Husk; Natural Rubber Latex | Copper; Nickel; Lead; Tin; Zinc; Aluminium | 13 |

Table 4 provides a brief description of these parameters and their values as proposed in (Grondona 1975). The authors used Grondona's proposed values of these parameters for the purpose of performing the simulation, which was simplified in several ways and based on the following assumptions:

- It considers only one grade of each commodity and is based on monthly data.
- It assumes that the CRD would have no stabilizing influence on commodity prices, and therefore overestimates the CRD's turnover, and so its direct effect on the money supply. (In reality, it can be expected that the CRD would have a significant stabilizing influence on some commodity prices at some times.)
- In addition, the CRD's overall scale is selected somewhat arbitrarily, and could readily be increased.

**Table 4.** Description of Parameters of Price Schedule.

| S. No | Parameters of Price Schedule | Description of Parameters | Grondona Suggested Values of Parameters |
|---|---|---|---|
| 1. | Range of Commodities | Only imported commodities should be included which are durable, essential and basic. Domestically produced commodities and fuel minerals are therefore not part of the system. | List of commodities depends on the country's primary imported commodities. |
| 2. | Initial Price index | The initial "Index" for each commodity is based on the average trend of previous years' average c.i.f.[1] price, (adjusted for inflation). | Average c.i.f. prices of each commodity of the country. |
| 3. | Width of Price Band | This is the difference between the CRD's lower and upper "points" for each commodity. This should depend on the normal range of fluctuations in each commodity's market price. | For initial simulation, the price band is set 10% below and above the initial "Index". |
| 4. | Block Size | 10% of average annual imports is used as a guideline to calculate the quantity in the "Block" of each commodity. | 10% of the country's average annual imports (in terms of quantity). |
| 5. | Size of Price Step between Successive Price Bands | This ratio should be fixed so that the upper and lower points in each price-band maintain a constant ratio. | The upper and lower points adjust by 5% of their initial levels, on withdrawal or receipt of each full Block. |

---

[1]    c.i.f. stands for Cost, Insurance and Freight.

## 4. Simulation Results

Initially, the authors computed the average inflation rate as the average of 2005–2008 annual Consumer Price Indices (CPI) to adjust Indonesia's annual primary imported commodity prices for inflation. Based on these average inflation-adjusted prices for the period of 2005–2008, the authors developed price schedules for all the primary commodities selected for simulation. After development of the price schedules, the authors performed simulations for each individual primary commodity for the period 2009–2018. The results of the simulations show that, on these assumptions, the Indonesian CRD would have accumulated reserves of Copper, Nickel, Lead, Zinc > 99%, Zinc < 99%, Aluminium, Coffee, Cotton, Sugar and Rice during the 10-year period. However it was found that the CRD's prices for some primary commodities (namely Tin, Cocoa beans and Rubber) were too low, so that the CRD purchased no reserves throughout the simulation period: consequently the initial Index of each of these commodities should have been set at a higher price (i.e., based on a different calculation than the other commodities).

The overall pattern of financial flows resulting from the Indonesian CRD's operations is to disburse Indonesian Rupiah abroad at times of falling commodity prices, and to withdraw Indonesian Rupiah from the domestic economy (which would otherwise flow abroad) at times of rising commodity prices. Such patterns are shown by the graphs in Figures 1 and 2 These flows would tend to reduce fluctuations in the prices paid by domestic users of commodities, in the prices received by foreign commodity producers, and secondarily in the demand for exports from Indonesia.

The financial effect of sales of reserves to the Indonesian CRD, (as shown in Figures 1 and 2) will be similar to an increase in government payments to the private sector. The Indonesian CRD's payments will appear first as an increase in banks' deposits at the Bank of Indonesia, which will influence the "reserve progress ratio of reserve deposits" and, if not counter-acted by the monetary authorities, the call rate. In this case it could lead to a further expansion of bank deposits by some multiple of the CRD's payments over following months.

Sales to the CRD will be made mainly by primary commodity exporters in foreign countries, and so the increase in the money supply will comprise first an increase in Rupiah bank accounts held by foreigners. Foreign-held Rupiah bank deposits may be used in various ways:

- They may be exchanged for national currency, in which case there could be some downward influence on the Rupiah exchange rate in relation to the currency in question, which is generally appropriate when commodity prices are falling (and so the value of the Rupiah in terms of those commodities is rising).
- They may be used to invest in Indonesian securities; and
- They may be used to purchase goods and services from Indonesia, in which case they could increase Indonesian exports.

Since the initial recipients of the CRD's Rupiah payments will be usual sellers of raw materials to Indonesia, the relative amounts of these three possibilities could be estimated to some extent from past statistics. In addition, to the extent that the Indonesian CRD's operations had the effect of maintaining Rupiah-denominated import prices of the commodities involved higher than the level to which they would have fallen in the CRD's absence, this will maintain the flow of commercial payments abroad above the level to which it would otherwise have fallen. Thus, the flow of Rupiah abroad resulting from the CRD's operations, and the commercial activities which this supports, can be expected to be larger than the amount disbursed by the CRD itself.

The financial flows resulting from purchases of reserves from the CRD will reduce bank deposits at the Bank of Indonesia, equivalent to receipts by the public sector. The fall in bank deposits at the Bank of Indonesia will alter the "reserve progress ratio of reserve deposits" and, if not counter-acted, the call-rate. In this case it could lead to a further reduction in bank deposits to some multiple of the CRD's transactions, which is generally appropriate when commodity prices are rising (and so the value of the Rupiah in terms of those commodities is falling).

Purchases from the CRD will generally be made by domestic users of the commodities concerned. That is, although the CRD will operate without national discrimination, Grondona proposed that the sites of its reserves should be chosen to be convenient for domestic users, who will as a result generally find the CRD's selling prices somewhat more attractive than foreign buyers, by the difference in cost of transport (Grondona 1975).

Purchases from the CRD will lead to a reduction in the flow of Rupiah abroad below what it would have been in its absence. In addition, imports by domestic users will be purchased at prices lower than they would have been in the CRD's absence, due to its influence in resisting rises in Rupiah-denominated import prices. Consequently, the reduction in the flow of Rupiah abroad resulting from the CRD's activities, and the commercial activities which this supports, will be larger than the value of purchases from the CRD itself.

Such effects, i.e., the increase/fall in the Indonesian CRD's reserves in response to a fall/rise in the market prices of the respective commodities, are evident from the graphs shown in Figure 1, and their corresponding effects on the financial flows of the Bank of Indonesia are also depicted in the graphs of Figure 2. For instance, in the graphs of Nickel in Figures 1 and 2, the Indonesian CRD accumulated reserves of Nickel during the first few months of 2009 due to a fall in Nickel prices, causing Rp 39,079 million expansion in the Indonesian money supply. And as a result of the rise in Nickel market prices during June–July 2009, it released the accumulated reserves of Nickel which caused a contraction of Rp 47,745 million in the Indonesian money supply.

It is important to note that the expansion of the Rupiah money supply was slightly less than the subsequent contraction, due to the sales premium, namely the difference between the buying and selling prices offered by the CRD. In the present case, Rp 8666 million is the sales premium earned by the Indonesian CRD during that period of the simulation. Grondona proposed that a portion of the sales premium should be used to cover the administrative and maintenance costs of the warehouses of the CRD, while any remainder could be transferred to a specified account which could be used for poverty alleviation or other programs for enhancing the welfare of the public within the country. However, continuing inflation in the host country will lead to loss of reserves again, even if world market prices are not rising. This can be partially compensated by Grondona's proposed remedy (described below). However, rapid inflation of about 10% per year or more would considerably reduce a CRD's beneficial influence, by preventing it obtaining reserves of some or all commodities.

Grondona suggested a solution to address this issue, namely that if the CRD obtains no reserves of a commodity for a certain specified period of time, say, two years (or more or less), then the initial index, and the upper and lower points for that particular commodity should automatically be increased by a certain stipulated percentage, say 5 percent (or other specified percentage) of the original initial levels. And the initial index and points should automatically be increased by the same percentage after each year until the CRD accumulates reserves of that commodity, the then index and points at which the CRD accumulated reserves becoming the new initial index and points. Grondona argued that such an adjustment to the initial index is required to tailor the CRD's gearing to the inflation prevailing in the country (Grondona 1975). Once again, the most appropriate pace of adjustment should probably not be uniform but should vary according to different commodities' conditions.

The Indonesian CRD stockpiles reserves of Zinc < 99% due to its initial gearing. Such an increase in Indonesian CRD reserves expands the Indonesian money supply by an amount equivalent to Rp 641,786 million (see the graphs in Figure 2). On the other hand, the purchase of Zinc < 99% Blocks by traders during August–December 2009 contracts the Indonesian money supply by Rp 70,849 million. Likewise, the Indonesian CRD had an expansionary effect on the money supply when Sugar market prices fell in January 2009, and the money supply contracted again in July 2009 due to a rise in Sugar market prices. Similar patterns were observed in the study described in (Ahmed 2015) of shorter simulations of four D-8 member states, which showed how each D-8 country's CRD stockpiled reserves of primary imported commodities (as a result of a drop in market prices of primary commodities), and released those reserves during subsequent periods of price hikes. Consequently, the transactions of each CRD

caused corresponding changes in each country's domestic money supply. That is, the national money supply of each D-8 country changed with the change in level of reserves of primary commodities stockpiled by their respective CRDs in response to an increase or decrease in market prices of the primary commodities. Such a mechanism helps to stabilise the prices and trade quantities of primary commodities and lessen the fluctuations in primary commodity markets during both slump and boom periods. As a result, the system would have exerted a corresponding stabilising influence on the real value of the Indonesian Rupiah.

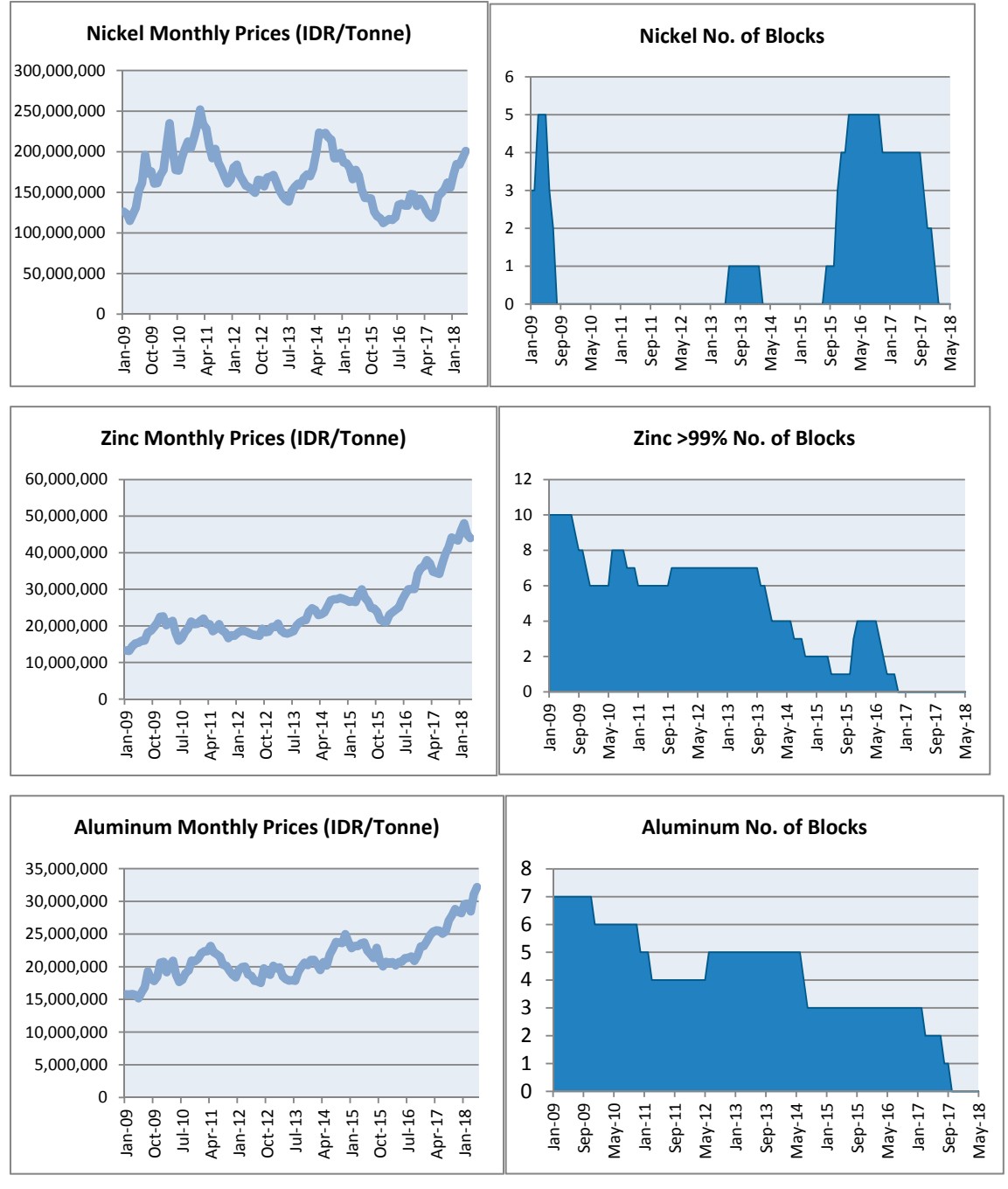

**Figure 1.** *Cont.*

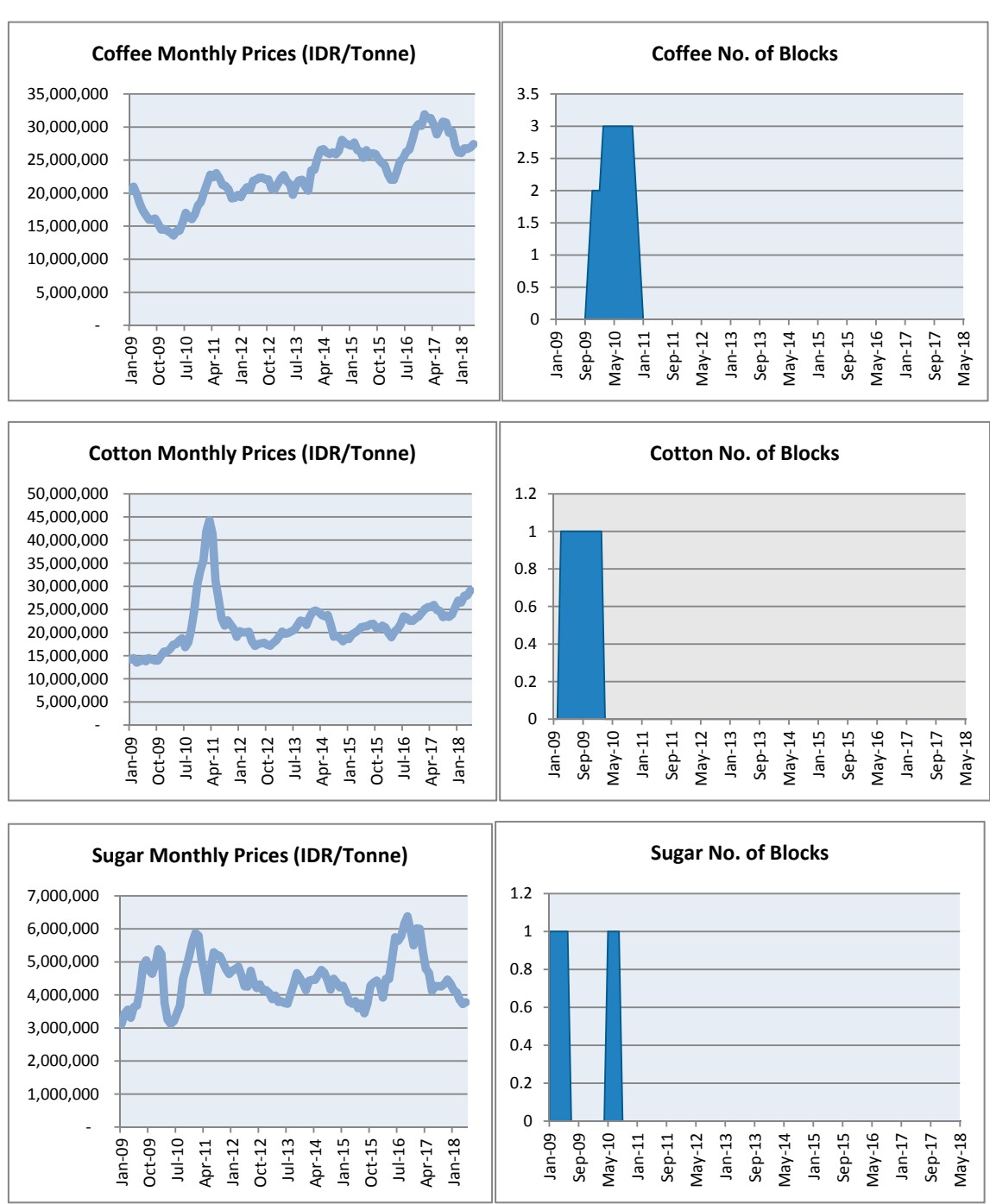

**Figure 1.** Changes in CRD Reserves of Various Commodities by the Indonesian CRD over the Period of 2009–2018.

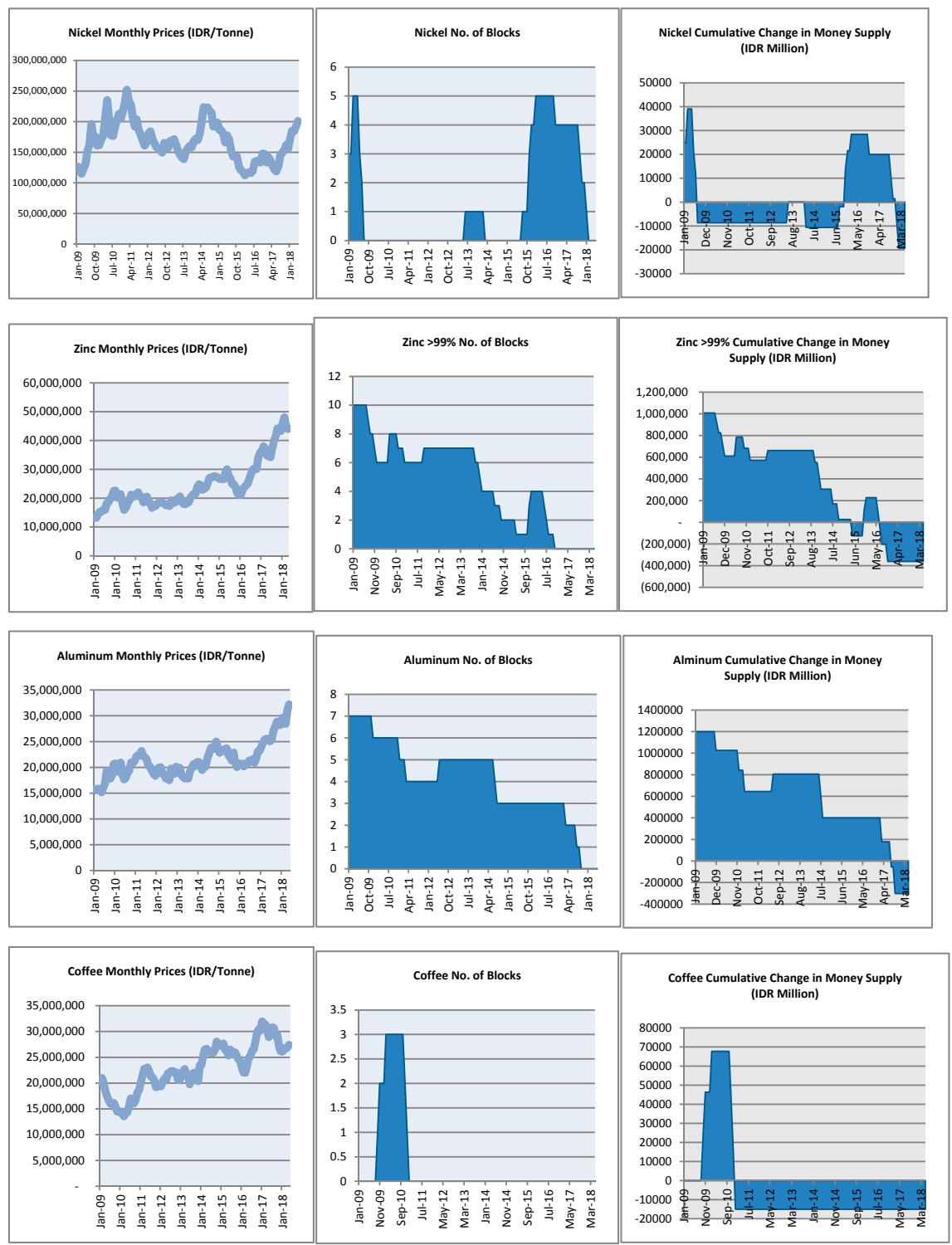

**Figure 2.** *Cont.*

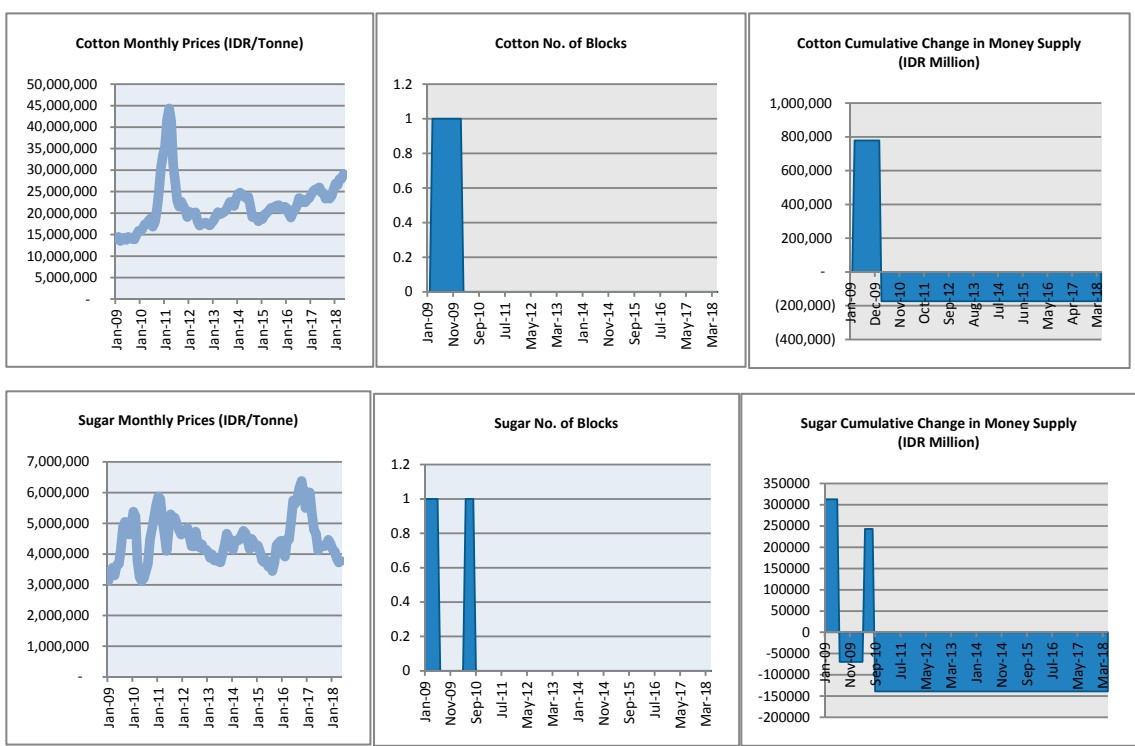

**Figure 2.** Financial flows caused by the Indonesian CRD Operations.

## 5. Discussion of Simulation

This simulation of the operation of an Indonesian CRD has shown the simplicity of simulating the operation of the Grondona system, under real (past) market conditions with high reliability. This is primarily due to its "automatic" (i.e., market price-driven) price-adjustment mechanism. This simplicity enabled this paper to analyze the probable result of its implementation in Indonesia in considerable detail, and with considerable confidence. That is, the results of the simulations clearly show the CRD's ability to stockpile reserves of primary imported commodities as a result of a drop in their market prices, and/or in response to a rise in the exchange-rate that reduces import prices, and to release those reserves during the following period of domestic price rises, whether due to market price rises or a fall in the exchange-rate (see graphs in Figure 1). Such a mechanism would help to stabilise Indonesia's trade quantities and Rupiah prices of primary commodities, and to lessen fluctuations in Indonesia's primary commodity trade over business cycles.

The simplicity of performing such useful simulations also shows how easy it would be to perform repeated runs with different initial values, in order to determine optimal conditions. This is one aspect of how this system is more practical than various proposals for international commodity-backed currency systems which unavoidably leave many important aspects to be decided by unpredictable negotiations, and/or unpredictable discretionary decisions during operation.

The simulation also demonstrated the impact of an Indonesian CRD's operations on the national money supply: the graphs in Figure 2 show the transactions of the Indonesian CRD causing corresponding changes in the domestic money supply. That is, the Indonesian CRD expanded the supply of Rupiah when there was a fall in the prices of primary commodities (and/or a rise in the exchange-rate) and contracted the supply of Rupiah when the prices of imported primary commodities rose (including in response to a fall in the exchange-rate). This pattern of financial flows clearly illustrates the counter-cyclical influence of the Grondona system, which improves the regularity and dependability of stock-holding of primary commodities over their price cycles.

The simulations also show that the uniform rule used to determine the initial conditions for all commodities was not optimal: the prices of some commodities never fell low enough for the CRD

to accumulate reserves, thereby limiting the CRD's stabilizing influence. In practice, the different conditions of each commodity would need to be considered in deciding the most appropriate details of the system's gearing for each. These include longer-term trends in each commodity's price, the quantities imported and used annually, relative inflation, and future prospects, including technological changes and other countries' imports.

*Uncertainties*

It is important also to consider three remaining uncertainties about the above simulations. First, the simulation is a "worst case", in that it was assumed that the CRD had no stabilising influence on world commodity market prices. In practice, the CRD might well have a stabilizing influence on prices of some commodities, particularly after a few years of operation, as discussed in (Collins 1985). The effect of this would be to reduce the quantities of purchases and sales by the CRD to some extent, thereby also reducing the size of changes in the Rupiah money supply. In one sense, this can be thought of as reducing "risk" to the Indonesian economy. However, since the timing of the movements will be strictly counter-cyclical for each commodity, larger movements are generally desirable, acting as an "automatic stabilizer" for the economy. Hence, to the extent that the CRD had a stabilising influence on world market prices, the government might decide to further increase the scale of the CRD.

It is also possible to simulate CRD operations on the assumption that it would have some significant stabilizing influence, although this introduces some uncertainty into the results. Nevertheless, the present "worst" case, which is the simplest and least uncertain to simulate, is useful for planning purposes. NB in practice it may not be "worst", since wide price fluctuations would generally be beneficial for the CRD, exerting a greater counter-cyclical stabilizing influence on the economy, while providing opportunities for greater profits from resale of greater quantities of reserves bought at lower prices. (NB Grondona recommended that the CRD should maintain a uniform margin between its buying and selling prices, rather than acting pro-cyclically in order to maximize its profits.)

A second uncertainty about establishing a CRD is the unavoidable uncertainty of future movements in commodity market prices. Remembering that market prices of many primary commodities show movements of −50% or more during recession and +100% or more during economic boom, the government would need to make arrangements that, in the event of a severe market "crash", it would be able to lease commercial storage space at short notice if reserves increased beyond the scale of normal price fluctuations for which it had prepared its own dedicated storage facilities.

Until implemented, a third uncertainty will remain about markets' likely reaction to the actions of a CRD during a currency crisis, in the form of extreme exchange-rate movements (including to unprecedented levels or at unprecedented speed). This is because, as the CRD's reserves increase, the money supply increases proportionately—which might be expected (on theoretical grounds) to reduce the value of the currency through markets' loss of confidence. However, at the same time, the real value of the currency, in terms of the commodities stored by the CRD, would be increasing proportionately as their prices fell, and reserves of the country's essential imports would be growing, thereby clearly improving the prospects for the economy in the future—as a result of which, by contrast, the currency might be expected to strengthen, thereby leading to even lower commodity prices. In advance of implementation it is not possible to know which of these sentiments will have the stronger influence on markets.

## 6. Conclusions: Contribution of a CRD to Preventing and Ameliorating Currency Crises

In summary, the most important result of the simulation described in this paper is to demonstrate a "practical" means of implementing a system of currency convertibility based on primary commodities. As discussed above, by making the terms of convertibility conditional, the maximum possible liability is limited to what the government of the implementing country can support independently. Moreover, by making the conditions of convertibility inversely dependent on the quantity of commodity reserves held, according to an openly published table, the system's operation is made "automatic" and so

both dependable and predictable, by both government itself and by market participants. By avoiding dependence on discretionary decisions which are essentially unpredictable, the operation of the system is made transparent, dependable, predictable and easy to simulate.

As described above, this concept was developed by Grondona, and is implemented in the above simulation by a "Commodities Reserve Department", or CRD. The simulation shows how the direct effects of a CRD's operations are exerted on commodity trade, by adding some reliably counter-cyclical stock-holding capacity to existing market conditions (any stabilizing effect that is achieved on commodity trade flows and prices being generally beneficial). In addition, the continuous operation of a CRD as a direct damping influence on fluctuations in commodity markets would introduce a counter-cyclical component into the growth of the money supply, and would have an indirect stabilizing influence on a range of other economic parameters, including the balance of payments, inflation, exchange-rate and interest-rates. As a consequence, in relation to currency crises, a CRD would play at least three roles of value to policy-makers.

### 6.1. Underlying Stabilising Influence

The CRD's continuous, underlying, stabilising influence could be expected to grow over the years following its implementation, as its role became familiar to market participants. Gradual recognition of the CRD's function as a modern equivalent of the gold standard would make a sudden major loss of confidence, as is needed to precipitate a currency crisis, progressively less likely as its successful operation continued.

### 6.2. Clear Guidance to Monetary Authorities

Movements in the CRD's reserves will provide a clear and undeniable, objective, public measure of changes in the real value of the currency in terms of the commodities handled, trends therein, and the effect of government policies. This will be of great help to governments in resisting pressure to distort monetary policy by expanding or contracting the money supply excessively. It will also be valuable for members of the general public trying to judge government policy. For example, inflationary policies would lead to steady reduction in the CRD's reserves of all commodities. A fortiori, permitting the reserves of all commodities to fall to zero, which would temporarily end the CRD's role of providing a measure of the minimum real value of the currency, would be widely seen as putting the currency and the economy at risk.

### 6.3. Concrete Step Towards Revival of Real Currency Convertibility

As a CRD's successful operation continued, there are numerous ways in which its stabilizing influence could be strengthened and extended to wider scale, for example by adjusting its "gearing", including other commodities or grades thereof, and in other ways. In addition, by initiating conditional currency convertibility, it will serve as a demonstration to other countries of the feasibility and potential benefits of following suit. Growing experience of the cumulative stabilising influence of several countries' CRDs could in turn lead to wider resumption of real convertibility.

#### 6.3.1. Further Work

For further work, it will be valuable to continue simulations of a CRD in Indonesia and other countries, using a variety of initial conditions different from the uniform rule used to date, so that the CRD accumulates reserves of all commodities. It will also be valuable to simulate the CRD with different values for the sizes of Blocks and other aspects of the "gearing" for different commodities in order to determine initial conditions for each country's CRD that are as nearly optimal as possible.

Simulations of several different countries' CRDs over the same time-scale are also potentially very valuable. This was performed for the first time for four D-8 member-states in (Ahmed 2015). Simulating how the combined monetary impact of several countries' CRDs' transactions, each in its own domestic currency, in response to changes in primary commodity market prices, could exert a cumulative

stabilizing effect over the business cycle, and in response to other external shocks is a promising extension of that work. Depending on the extent of overlap in the commodities handled, even without any formal coordination, the different CRDs' operations would tend to expand and contract the supply of the different CRDs' currencies in synchrony, thereby having a mutually stabilizing influence on their exchange-rates. This is a highly desirable outcome, and hence could be an important topic for joint investigation by D-8 countries. In particular, because the influence of each country's CRD acting alone would be relatively small relative to world commodity markets, the simultaneous operation of several different CRDs would have much greater influence. In addition, any effect of synchronising changes in the different countries' money supplies would exert some influence towards creating an informal currency "bloc".

### 6.3.2. Potential for D-8 Countries' Initiative

Considering the ongoing risk of financial crises around the globe, both "natural" due to the instability in world financial markets, and politically induced by US government policies, this study has potentially important implications for policy makers in the D-8 member-states. As discussed elsewhere, the Grondona system has been recognised as Shariah-compliant, and the above simulation shows that it could contribute to improving D-8 member-states' macro-economic stability. Consequently, implementation of the Grondona system by the leading D-8 member-states could be a preliminary step towards insulating themselves from economic instability, and thereby spreading their monetary and trade stabilising influence on other OIC member-states and beyond. In this case the potential benefits of such an initiative for the implementing countries include the following:

- It would partially stabilise the real value of the participating currencies, albeit within a flexible range of approximately ±10%.
- Its operations would lead to counter-cyclical changes in each currency's money supply, tending to maintain a constant real value of each.
- It would have direct stabilising effects on domestic prices of primary commodities. By this, it not only helps to preserve economic stability but also to reduce the economic dependence of developing OIC countries on developed countries for financial assistance.
- It will help governments to resist external political pressures. As an example, deliberate destabilisation of the exchange-rate would lead to wider movements in domestic commodity prices, which would increase the premiums earned by the CRD over the full cycle, while making stronger stabilising changes in the money supply. Hence, for example, a country targeted for destabilisation, as during the South East Asian currency crisis, would be better able to withstand such political pressure with a CRD in operation than without.

**Author Contributions:** Conceptualization, P.C.; Methodology and computation, J.A.; Drafting, P.C. & J.A.; Supervision, A.K.M.

**Funding:** This research received no external funding.

**Acknowledgments:** We thank Saba Raja for invaluable assistance in performing the simulations for Indonesia.

**Conflicts of Interest:** The authors declare no conflict of interest.

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
