# Peer review of "Simulation of the Grondona System of Conditional Currency Convertibility Based on Primary Commodities, Considered as a Means to Resist Currency Crises"

_jrfm, doi:10.3390/jrfm12020075_

Round 1

Reviewer 1 Report

I have several serious concerns regarding your paper:

1

Academic analysis requires an approach based on facts, evidence and arguments. Instead, your paper reads like a political pamphlet at times (e.g., consider p. 2 of the Introduction). 

2

There is an abundant literature on currency crisis comprising various generations of currency crisis models. These models explain why crises erupt and point to political remedies. Your work totally ignores this rich literature and deals with the symptoms instead.

3

Your work totally ignores how modern economies operate. There is no discussion of the potential drawbacks and the economics costs of your proposal. 

Author Response

Thank you for your comments, which have been very helpful for improving the paper.

Please find our responses in the attachment. We also attached a copy of the edited paper  at the end of the responses, with new and/or much-altered material in red.

I have also made many other minor changes which are not in red, but which are intended to improve clarity and/or the tone of the paper.

I hope you will consider these changes satisfactory.

Sincerely,

                 Patrick Collins

Reviewer 2 Report

The paper focuses on the Gordona system of conditional currency convertibility in Indonesia. The authors simulate the operation of the system by following the theory proposed in the literature.

The results of simulations prove that commodity price movements generate automatic counter-cyclical stock-holding of the system which causes monetary flows that resist the changes and so the system protects from currency crises.

The paper is interesting because it shows how a model existing in the literature could protect a country from currency crises if implemented. However, its quality of presentation is very low and it is not well-organized, namely the article has not got the structure of a scientific work. 

Below we suggest some modifications:

1) the Introduction provides a good framework of the currency crises. However, the authors should highlight the contribution of the paper to the existing literature. Furthermore, the outline of the paper is completely missing.

2) Section 2 could be included in the Introduction and could be extended in order to describe better the recent literature and also to compare the recent literature with the research approach of the authors.

3)The section of conclusion is missing. It is important to summarize the main results and contribution of the paper.

Author Response

(The authors gave the same response as above.)

Reviewer 3 Report

This is an interesting and original article,which could be improved by clarifying several points that at the moment remain obscure. Moreover, there are a few theoretical issues that could be addressed in more detail and cannot be solved buy a simple simulation. 

In page 1: what is the exact issue at hand? the paper proposes the use of the Grondona system to stabilise the Indonesian rupiah based on a basket of commodities that Indonesia trades regularly. This needs to be mentioned in the very beginning of the article. 

Page 4, paragraph around line 140. What is the "shariah perspective" on currencies and what makes a system compliant with it? 

It is unclear to me why this system is superior to the gold standard. Yes, it is explained that unlike the gold standard the Grondona mechanism is counter-cyclical, but it does not address other problems of the gold standard that this system seems to replicate, as far as I can understand. These oter weaknesses are barely mentioned.  

It seems every country can implement the Grondona mechanism in relation to its own currency, so in principle in relation also to the commodities that each country trades regularly.

However, “the system would be implemented by a Commodities Reserve Department of the government” (page 4, line 190). But how would these CRD be coordinated? they would indirectly interact via the global commodity markets, right? 

The proposal of the Grondona mechanism in Indonesia assumes that there is a market for aluminum that is at least as strong as the market for currency. Otherwise, the prices of the commodity would fluctuate differently in relation to the other more and less demanded currencies by the buyers of aluminum in those currencies.  

On choice of commodity to stabilize the currency: where does the aluminum to be used for the stabilization of the currency come from? Is it state-owned? Is it Indonesian at all? In page 8 around line 260 “supplying countries” are mentioned. 

In page 10, line 315 it is admitted that the CRD would have an impact on the commodity prices. In the discussion this point is barely addressed and I find a key issue. I suggest the authors consider expanding on this point.

The commodities in Table 2 are mentioned as imports from Indonesia. Is this correct? Should export commodities not be preferred? 

In view of my questions above, perhaps the authors could clarify the feasibility of the Grondona mechanism. If the CDR had the commodities already and was to stabilize the currency by selling the commodities (like Chile did with state-owned copper exports) the idea would make a lot of sense. But if the commodity still has to be bought or printed, then the prices of the commodities may be affected before they could be used to stabilize the currency. 

In general, I wonder how commodity prices would be affected by using them as stabilizers. As mentioned in the paper (page 18, line 500) commodity prices are extremely volatile. So could they give stability to currencies, if their prices are themselves so unstable? 

The costs of stockpiling are not addressed either.

Discussion. I can see why the authors may think that a Grondona mechanism is a good idea. However, I am not sure I would define it as practical (underlined in original text)in its implementation. An independent CDR is still to be created and the commodities still need to be obtained before the Grondona mechanism can actually be used to stabilize the Indonesian currency. What would be the impact and implications of the implementation process? 

The paper also abstracts from the existence of other currencies, other actors and commodities in the world against which the prices of other commodities vary. Why would it not lead to competition among CRD of different countries and a race for commodities and currencies to the bottom?

Author Response

(The authors gave the same response as above.)

Round 2

Reviewer 2 Report

The author have addressed all my comments. They have improved the structure and organization of the paper.